# Overexpression of UBA5 in Cells Mimics the Phenotype of Cells Lacking UBA5

**DOI:** 10.3390/ijms23137445

**Published:** 2022-07-04

**Authors:** Sujata Kumari, Sayanika Banerjee, Manoj Kumar, Arata Hayashi, Balakrishnan Solaimuthu, Einav Cohen-Kfir, Yoav D. Shaul, Alexander Rouvinski, Reuven Wiener

**Affiliations:** 1Department of Biochemistry and Molecular Biology, The Institute for Medical Research Israel-Canada, Hebrew University-Hadassah Medical School, Jerusalem 91120, Israel; sujatasuman2007@gmail.com (S.K.); sayanika.banerjee@gmail.com (S.B.); manoj.medchem@gmail.com (M.K.); arata.hayashi@mail.huji.ac.il (A.H.); solaimut.balakrishna@mail.huji.ac.il (B.S.); einavck@walla.co.il (E.C.-K.); yoavsh@ekmd.huji.ac.il (Y.D.S.); 2Department of Microbiology and Molecular Genetics, The Institute for Medical Research Israel-Canada, The Kuvin Center for the Study of Infectious and Tropical Diseases, Hebrew University-Hadassah Medical School, Jerusalem 91120, Israel; alexander.rouvinski@mail.huji.ac.il

**Keywords:** ufmylation, UBA5, UFC1, E1-activating enzymes, E2-conjugating enzymes, ubiquitin-like protein, UFM1

## Abstract

Ufmylation is a posttranslational modification in which the modifier UFM1 is attached to target proteins. This conjugation requires the concerted work of three enzymes named UBA5, UFC1, and UFL1. Initially, UBA5 activates UFM1 in a process that ends with UFM1 attached to UBA5’s active site Cys. Then, in a trans-thiolation reaction, UFM1 is transferred from UBA5 to UFC1, forming a thioester bond with the latter. Finally, with the help of UFL1, UFM1 is transferred to the final destination—a lysine residue on a target protein. Therefore, not surprisingly, deletion of one of these enzymes abrogates the conjugation process. However, how overexpression of these enzymes affects this process is not yet clear. Here we found, unexpectedly, that overexpression of UBA5, but not UFC1, damages the ability of cells to migrate, in a similar way to cells lacking UBA5 or UFC1. At the mechanistic level, we found that overexpression of UBA5 reverses the trans-thiolation reaction, thereby leading to a back transfer of UFM1 from UFC1 to UBA5. This, as seen in cells lacking UBA5, reduces the level of charged UFC1 and therefore harms the conjugation process. In contrast, co-expression of UBA5 with UFM1 abolishes this effect, suggesting that the reverse transfer of UFM1 from UFC1 to UBA5 depends on the level of free UFM1. Overall, our results propose that the cellular expression level of the UFM1 conjugation enzymes has to be tightly regulated to ensure the proper directionality of UFM1 transfer.

## 1. Introduction

Modifications of proteins by ubiquitin or ubiquitin-like proteins (UBLs (play a role in a wide spectrum of cellular processes ranging from proliferation to death [1,2,3,4]. These modifiers are conjugated to target proteins via a three-enzyme cascade involving E1, E2 and E3 groups of enzymes, whereby each group has its own distinct role in the conjugation machinery [5,6,7]. Initially, E1, in an ATP-dependent manner, hydrolyzes ATP and forms a thioester bond with Ub/UBL C-terminal Gly and its active site Cys [8]. Then, the modifier is transferred from the E1 to the E2, forming a thioester bond with the latter [9]. Finally, with the help of an E3 enzyme, the modifier is transferred to the target protein, forming an isopeptide bond with the latter [10]. To date, two E1 enzymes, tens of E2s and hundreds of E3 enzymes are known to function with ubiquitin, allowing the conjugation of ubiquitin to a wide spectrum of targets. In contrast, the number of enzymes that play a role in UBLs’ conjugation is significantly lower, although each UBL has its own E1, E2 and E3 enzymes [6].

UFM1 (ubiquitin fold modifier 1) is a UBL that was shown recently to be involved in key cellular processes, including protein translation, unfolded protein response (UPR), and DNA damage response [11,12,13,14,15,16]. Conjugation of UFM1 to target proteins is executed by the function of the E1, UBA5, the E2, UFC1 and the E3, UFL1 [17,18]. UBA5 belongs to the family of non-canonical E1 enzymes. It includes an adenylation domain that harbors the active site Cys (C250), a UFM1-interacting sequence (UIS) that binds non-covalently to UFM1, and a UFC1-binding site (UBS) that is responsible for the interaction with UFC1 [19,20,21]. UFM1 is activated by the dimeric UBA5 in a unique trans-binding mechanism in which UFM1 binds the UIS of one protomer and is transferred to the active site Cys of the other protomer, forming a thioester bond [22,23,24]. Then, a trans-thiolation reaction occurs whereby UFM1 is transferred from UBA5 to UFC1’s active site Cys [25,26]. Finally, with the help of UFL1, UFM1 is transferred to Lys residue on a target protein, forming an isopeptide bond [18].

In this work, we interfered with the UFM1 machinery by silencing UBA5 or UFC1 or by overexpressing them, in human cell lines. We then tested how these manipulations affect cell migration, a cellular process that has been shown to involve protein modification by UFM1 [27,28]. As expected, deletion of UBA5 or UFC1 led to a similar phenotype of reduced cell migration. However, while overexpression of UBA5 reduced cell migration, overexpression of UFC1 had no effect or slightly increased migration. In order to understand why overexpression and silencing of UBA5 yield similar phenotypes, we determined the level of charged UFC1 in these cells. Both manipulations (UBA5 deletion or overexpression) led to a reduction in the level of charged UFC1, suggesting a similar impact on the ufmylation process. Overexpression of UBA5 enforced back transfer of UFM1 from UFC1 to UBA5, thereby reducing the level of charged UFC1. However, overexpression of UBA5 together with UFM1 prevented this back transfer and accordingly abrogated the phenotype seen with overexpression of UBA5 alone. Overall, our results suggest that the expression level of the conjugating enzymes in the cell is highly regulated to ensure proper ufmylation, and accordingly any deviation from these levels could be devastating.

## 2. Results

### 2.1. Alterations in UBA5 Expression Level Affect Cell Migration

To study whether ufmylation is involved in the ability of cells to migrate, we generated HeLa cells with knockdown (KD) of Uba5 or Ufc1, and HEK293T cells with knockout (KO) of these genes (Figure 1A). In addition, we overexpressed EGFP-UBA5 or EGFP-UFC1 in HeLa cells (Figure 1B). Then, with these cells in hand, we tested the ability of the HeLa cells to migrate in a Boyden chamber-based trans-well migration assay [29]. As shown in Figure 1C, UBA5 KD and UFC1 KD significantly reduced the ability of the cells to migrate compared to the control cells. Interestingly, UBA5 KD did not abolish cell migration as was observed in UFC1 KD cells, suggesting that there is still residual UBA5 activity in these cells, or alternatively that another E1 compensates for the lack of UBA5. Unexpectedly, while overexpression of UFC1 enhanced cell migration (Figure 1D), overexpression of UBA5 significantly reduced it (Figure 1E). This reduction was dependent on the catalytic activity of UBA5, since overexpression of EGFP-UBA5 with a mutation in the active site (C250A) did not affect cell migration (Figure 1E). Overall, our results show that overexpression of catalytically active UBA5, but not UFC1, mimics the effect of non-functional UFM1 machinery. 

### 2.2. Overexpression of UBA5 Reduces the Level of Charged UFC1 in the Cell

UBA5 activates UFM1 and then transfers the latter to UFC1, forming charged UFC1. Therefore, as expected, western blot analysis of UBA5 KD HeLa cells or UBA5 KO HEK 293T cells showed no charged UFC1 (Figure 2A). However, it was not clear whether overexpression of UBA5 affected the level of charged UFC1. Therefore, we overexpressed UBA5 in HeLa or HEK293T cells and tested the level of endogenous charged UFC1. As shown in Figure 2B–D, overexpression of UBA5 WT, but not C250A, reduced the level of charged UFC1 similarly to cells lacking UBA5.

While charging of UBA5 with UFM1 requires energy (i.e., ATP hydrolysis), transfer of UFM1 from UBA5 to UFC1 does not. In this trans-thiolation reaction, one thioester bond (UBA5~UFM1) is broken and another one (UFC1~UFM1) is formed; thereby, no net investment of energy is required. This implies that transfer can go in both directions, which are energetically equivalent. Thus, the direction of the reaction will be determined by the concentrations of UBA5, UFC1 and free UFM1. Therefore, if UBA5 is overexpressed and the amount of free UFM1 cannot satisfy charging of UBA5, the latter can be charged in a reverse transfer, where UFM1 is taken back from charged UFC1, resulting in a reduction in the charged UFC1 level. In such a scenario, excess UBA5 drains the UFM1 pool. However, if UBA5 is overexpressed together with UFM1, one would expect that a reduction in charged UFC1 levels will not be observed. Indeed, as shown in Figure 2E, overexpression of UFM1 together with UBA5 prevents the latter decreasing the level of charged UFC1. Overall, our results show that the charged UFC1 level decreases upon UBA5 overexpression, presumably due to the back transfer of UFM1 from UFC1 to UBA5. 

To further support that the reduction in charged UFC1 upon overexpression of UBA5 is due to the lack of free UFM1, we established an in vitro system using pure UBA5, UFC1 and UFM1 proteins. Using this system, we tested charging of UFC1 with UFM1 at different concentrations of UBA5. In these experiments, the concentrations of UFC1 and UFM1 were kept constant at 1 µM, while the UBA5 concentration varied from 0.1 µM to 15 µM. As expected, incremental increases in the concentration of UBA5 resulted in reduced amounts of charged UFC1, and accordingly more uncharged UFC1 was detected (Figure 2F). Specifically, while at low UBA5 concentration (0.1 µM) only 20% of the total UFC1 was in the uncharged form, at a high UBA5 concentration (15 µM) 80% was in that form (Figure 2G). Interestingly, at UBA5 concentrations above 1 µM, a band that resembles charged UBA5 appears at zero-time points, although no ATP was added (Figure 2F). This band appears only in samples having UBA5 and UFM1 (Appendix A). This suggests that the ATP that comes from the *E. coli* and is retained during the UBA5 purification steps allows the charging. Taken together, our data support the conclusion that increasing the concentration of UBA5 without elevating UFM1 concentration can reduce the level of charged UFC1.

### 2.3. Cells Express Less UBA5 Than UFM1 or UFC1

Our finding that overexpression of UBA5 reverses the trans-thiolation reaction prompted us to determine the levels of endogenous UFM1, UBA5 and UFC1. To that end, we performed western blot analysis for these proteins from HEK293T cells. Then, using recombinant proteins of UBA5, UFC1 and UFM1, at known concentrations, we determined the levels of these proteins in the cell. As shown in Figure 3A,B, we found that the expression levels of UFC1 and UFM1 are respectively 3-fold and 5-fold higher than that of UBA5. Overall, these results support our finding that upon overexpression of UBA5, the endogenous UFM1 will not be able to satisfy full charging of UBA5, and back transfer of UFM1 from UFC1 to UBA5 may take place.

Next, we assessed the expression level of these genes in different human cells. We used RNAseq data from patients provided by cBioPortal, and calculated the average RNA expression level of the above genes from 30 different tissues (see Appendix A). As shown in Figure 3C, the RNA expression level of UFC1 is 2.5-fold higher than that of UBA5, and the UFM1 expression level is between that of UFC1 and UBA5. Overall, these results are in line with our protein data, suggesting that overexpression of UBA5 will drain the free UFM1 and accordingly will reverse the ufmylation reaction. 

## 3. Discussion

Common to all the modifications by Ub or UBLs is the requirement for functional E1 and E2 enzymes, and, not surprisingly, deletion of these enzymes prevents the conjugation process. Here, we showed that in the UFM1 system, overexpression of the E1, UBA5, mimics the phenotype of UBA5 deletion. This, we suggest, is due to back transfer of UFM1 from UFC1 to UBA5. Alternatively, overexpressed UBA5 interacts with UFM1 in a nonproductive manner, thereby titrating out UFM1 and accordingly reducing the level of charged UFC1. However, since overexpression of UBA5 C250A does not affect the level of charged UFC1, the possibility of titrating out UFM1 is less likely. Our results, therefore, imply that if one is interested in silencing ufmylation, elevating the expression level of UBA5 can be exploited as a simple strategy for reaching this goal. 

UFM1, like ubiquitin and other UBLs, is formed as a precursor that undergoes cleavage at the C-terminus, yielding the mature form [30]. In the case of UFM1, this process includes the removal of the last two amino acids, Ser-Cys, exposing the C-terminal Gly. Currently, we cannot differentiate between the two forms of UFM1, suggesting that the level of functional mature UFM1 could be less than what we concluded based on our western blot analysis. Moreover, it is not yet clear whether the immature form of UFM1, which cannot be activated by UBA5, can intervene with the ufmylation process by forming noncovalent interactions with the conjugating enzymes. 

Our study focused on the trans-thiolation reaction that occurs in the transfer of UFM1 from its E1 to E2 and on the reversibility of this reaction. Similar trans-thiolation reactions happen in the transfer of Ub from E2s to HECT or RBR E3 ligases [10]. These E3 ligases harbor an active site Cys that accepts the Ub and forms thioester bonds with the latter prior to the transfer to the target protein. This raises the possibility that in E3 ligases that form thioester bonds with Ub/Ubl, a certain expression level of these E3s relative to their cognate E2s is essential to prevent back transfer from E3 to E2. Currently, the mechanism of the UFM1 E3 ligase, UFL1, is not clear, and how its expression level affects ufmylation is still under investigation. 

The UFM1 machinery possesses a single E1 and a single E2. In contrast, Ub has ~40 E2 enzymes that function with two E1 enzymes. This implies that overexpression of Ub E1 can lead to a complex response that will have different effects on the E2s. In this case, some E2s will be more potent than others for the back transfer, which in turn will selectively alter the cellular ubiquitination pattern. Overall, while E1 overexpression can be used for silencing ufmylation, in the case of Ub and other Ubls, this is more complex and thereby less attractive for silencing purposes. 

Parallel to our finding that overexpression of UBA5 halts the ufmylation machinery, we found that UFM1 plays a role in cell migration. Interestingly, while our data show that prevention of ufmylation in HeLa cells reduces the ability of the cells to migrate, in gastric cancer cells, downregulation of UFM1 expression enhances the migration and invasion of these cells [27]. To date, the mechanisms by which ufmylation affects cell migration are unknown, and detailed study is required. In addition, the connection of ufmylation to cell migration, which is a key process in metastases development, strengthens the potential embedded in this system as a target for drug development. 

## 4. Materials and Methods

### 4.1. Cells Culture and Growth Condition

HeLa and HEK293T cells were maintained in high glucose DMEM supplemented with 10% fetal bovine serum, 2 mM l-glutamine, 0.1 mg/mL penicillin and 0.1 mg/mL streptomycin. Maintained cells were examined for mycoplasma using EZ-PCR^TM^ Mycoplasma Detection Kit (Biological Industries, Beit HaEmek, Israel). Cells were grown at 37 °C in a humidified atmosphere of 5% CO_2_ and 95% air.

### 4.2. Generation of UBA5 and UFC1 Knockdown Cells

HEK293T cells were grown at 90% confluency and transfected with psPAX2, pMD2.G, and sh scramble (TRC1/1.5 Sigma Aldrich) or shUBA5 (08018 Sigma Aldrich, St. Louis, MO, USA) or shUFC1 (004172 Sigma Aldrich) using Transporter ^TM^ 5 transfection reagent. At 6 h post transfection, medium was changed to fresh medium, and at 24 h and 48 h post transfection, the medium was collected. The medium was centrifuged at 4200× *g*, 4 °C for 30 min, and supernatant was collected. The supernatant was concentrated through 1 mL 20% sucrose solution in PBS and ultra-centrifuged at 38,000 rpm, 4 °C for 3 h. Pellets were re-suspended in 70 µL PBS (*w*/*o* Ca^2+^ and Mg^2+^) and stored at −80 °C until use. These lentivirus suspensions were titered on HEK293T monolayers seeded in a 96-well plate using 1:10 dilutions. Infected cells were selected in 2 µg/mL puromycin medium, and concentrations of lentiviruses were calculated. Using MOI-10 (Multiplicity of Infection), HeLa cells were infected with lentiviruses containing shRNA-scramble or shRNA-UBA5 or shRNA-UFC1 and incubated for 2 h, before washing them twice with fresh medium. At 24 h post infection, fresh medium containing 2 µg/mL puromycin was added. Knockdown of UBA5 or UFC1 was verified by DNA sequencing, qRT-PCR and western blot.

### 4.3. Generation of UBA5 and UFC1 Knockout Cells

Guide RNA (gRNA) sequences for CRISPR /Cas9 were designed at the CRISPR design website (http://crispr.mit.edu accessed on 11 October 2018). The insert oligonucleotide for human UBA5gRNA was 5′-AAGCAGCAGAACATACT CTG-3′ and UFC1gRNA was 5′-TTGTGGGTGCAGCGACTGA-3′. The UBA5 and UFC1 gRNA targeted exon1 of the UBA5 and UFC1 genes. The complementary oligonucleotides for gRNAs were annealed and cloned into lenti crispr V2 vector (Add gene #52961). HEK293T cells were transfected with either lenti crispr v2 UBA5g RNA or lenti crispr v2 UFC1g RNA using Transporter^TM^ 5 reagent, as above. Lentiviruses were purified, titered and concentrated. HEK293T cells were infected with the lentiviruses (MOI-10) containing UBA5gRNA or UFC1gRNA and selected with 2 µg/mL puromycin medium. Knockout of UBA5 or UFC1 was verified by DNA sequencing and western blot.

### 4.4. Cloning and Mutagenesis

Human UBA5 and UFC1 were subcloned into mammalian expression vector pEGFP-C1 by restriction digestion and ligation reactions using routine cloning methods. UBA5 (C250A) and UFC1 (C116A) were generated by Pfu Ultra II Fusion HS DNA polymerase according to manufacturer’s protocol (Agilent, Santa Clara, CA, USA). The UFM1-pEGFP-C1 construct was generated using Gibson assembly (Gibson assembly master mix, New England Biolabs, Ipswich, MA, USA). All the constructs were verified by DNA sequencing. 

### 4.5. Western Blotting

Cells were lysed in Bis-Tris lysis buffer (50 mM Bis-Tris pH-6.5, 100 mM NaCl, 1% NP-40) containing protease inhibitor cocktail set III, EDTA-free (Calbiochem cat. No. 539134) at a dilution of 1:200. Total protein concentration was analyzed using Bio-Rad’s Protein Assay Dye Reagent Concentrate (Cat# 5000006). A total of 20 µg of total protein from each cell lysate was separated by 12% Bis-Tris SDS-PAGE and transferred to a PVDF membrane. The primary antibodies UBA5-ab131128 (1:1000, Abcam, Cambridge, UK), UFC1-ab189252 (1:10,000, Abcam), GAPDH (1:500, Santa Cruz, Dallas, TX, USA), UFM1-ab109305 (1:1000, Abcam), and the secondary antibodies HRP-conjugated goat anti-mouse and goat anti-rabbit (H + L) (1:10,000, Jackson Laboratories, Bar Harbor, ME, USA) were used. ImageJ was used for western blot quantifications. 

### 4.6. Fluorescent Gel Imaging

HeLa cells were transfected with EGFP, EGFP-UBA5, EGFP-UBA5 (C250A), EGFP-UFC1 and EGFP-UFC1(C116A) plasmid and Transporter 5 reagent separately. At 24 h after transfection, cells were harvested in Bis-Tris lysis buffer. Total protein concentration of each cell lysate was measured as above. A total of 20 µg of protein for each cell lysate was loaded on a 12% Bis-Tris SDS-PAGE gel with 1× sample buffer with and without β-mercaptoethanol (BME; Sigma Aldrich M6250). Gels were imaged using the ChemiDoc MP Imaging system (Bio-Rad, Hercules, CA, USA).

### 4.7. Trans-Well Migration Assay

We performed the cell migration assay with HeLa cells since they are larger in size and are not as easily detached from the substratum of the seeded chamber as the HEK293T cells. Cell migration was assayed using a 24-well trans well chamber (Corning Costar, Corning, NY, USA) with an 8.0 μm pore size. A total of 32,000 cells were suspended in serum-free DMEM high glucose medium and seeded onto the upper chamber. Then, medium containing 10% FBS was added to the lower chamber. After 24 h culture, the nonmigratory cells were removed from upper chamber. The migratory cells in the lower surface were fixed in 4% para-formaldehyde (PFA), incubated with methanol for 20 min, stained with 0.1% crystal violet and photographed under a light microscope (Nikon ECLIPSE Ts2R, Tokyo, Japan).

### 4.8. In Vitro Thioester Assay

Recombinant UFC1, UBA5 and UFM1 were purified from *E. coli* as previously described [23]. For activity assay, we incubated UFM1 (1 μM) and UFC1 (1 μM), together with UBA5 (concentrations are indicated in the figure) in a buffer containing Bis-Tris (50 mM pH 6.5), NaCl (100 mM) and MgCl_2_ (10 mM) at 30 °C. The zero-time point was collected before ATP (5 mM) was added. The samples were collected after 30 min. Both 0 and 30 min time points were quenched using SDS sample buffer without BME. Finally, samples were loaded on 12% Bis-Tris SDS-PAGE and visualized by InstantBlue Coomassie Protein Stain (ab119211; Abcam) staining. UFC1 band quantification was performed using imageJ.

### 4.9. UBA5, UFC1 and UFM1 Quantification Assay

HEK293T cells were seeded in 10 cm plates at a density of 2.5 × 10^6^ a day before the experiment. Cells were harvested in 500 µL lysis buffer (50 mM Bis-Tris pH-6.5, 100 mM NaCl, 1% NP-40) containing protease inhibitor cocktail set III, at a dilution of 1:200, sonicated and centrifuged at 15,000 rpm, for 15 min at 4 °C. Supernatant was collected to new tubes and total protein concentration was measured. Total protein amounts of 8 µg, 16 µg, 24 µg and 32 µg were loaded on SDS-PAGE (4–20% Mini-PROTEAN^®^ TGX™ Precast Protein Gels, 10-well #4561094) and transferred to PVDF membrane. The amount of proteins were visualized after immunoblotting using anti UBA5, anti UFC1 and anti UFM1 antibody. Each band was quantified using ImageJ. To transform band intensity to protein amount, we used known amounts of pure recombinant UBA5, UFC1 and UFM1 that enabled us to generate a calibration curve. Then, using this curve, we determined the amount of protein in the cell lysates. Data are shown as nanomole of indicated protein divided by total protein (presented in grams).

### 4.10. RNAseq Data Analysis

Gene expression data from TCGA PanCancer Atlas for the indicated genes were obtained from the cBio Cancer Genomics Portal (http://cbioportal.org, accessed on 22 February 2022). Mean mRNA level for each gene from 30 different cancerous cells were calculated. Then, data were visualized as box plot with Wilcoxon signed rank test *P* values (Figure 3C) or bar plot using R package ggplot2 [31] (Appendix A).

## Figures and Tables

**Figure 1 ijms-23-07445-f001:**
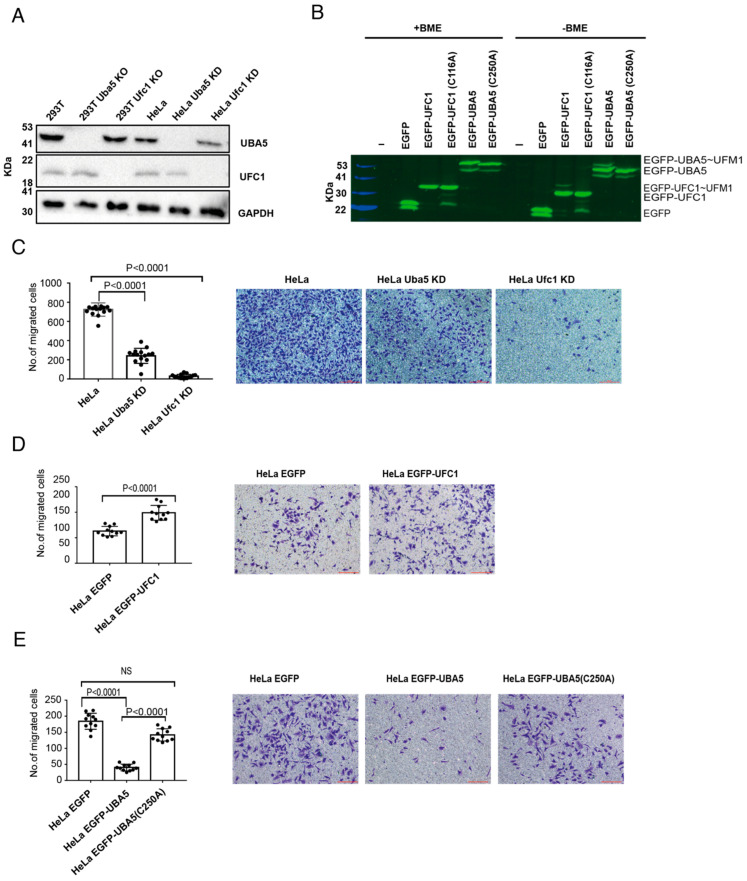
**Alteration of UBA5 expression level reduces cell migration:** (**A**) Western blot analysis of UBA5 and UFC1 levels in the indicated cell lines. GAPDH was used for loading control. (**B**) Fluorescent SDS-PAGE showing the expression level of EGFP-UBA5 or EGFP-UFC1 (WT or mutant) in HeLa cells. EGFP and EGFP-UBA5 both run as two bands in the SDS-PAGE. β-mercaptoethanol (BME) is a reducing agent that breaks thioester bonds. In the absence of BME, EGFP-UBA5 forms a third band corresponding to EGFP-UBA5~UFM1. In the absence of BME, EGFP-UFC1 forms two bands corresponding to uncharged and charged EGFP-UFC1. (**C**) Trans-well migration assay of cells lacking UBA5 or UFC1. The data are shown as the number of migrated cells; each value represents the mean ± SD. The *p* value was determined by Student’s *t* test. Representative images of the indicated migrated cells; scale bar 100 µm (right). (**D**) Similar to (**C**), but with cells overexpressing EGFP or EGFP-UFC1. (**E**) Similar to (**C**), but with cells overexpressing EGFP or EGFP-UBA5 (WT or mutant).

**Figure 2 ijms-23-07445-f002:**
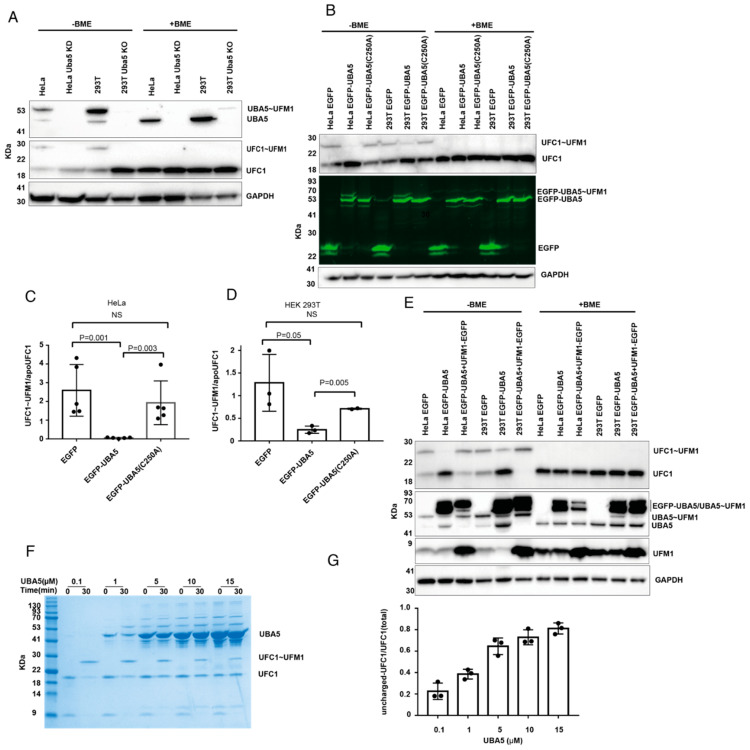
**Interference with UBA5 level affects the charged level of UFC1.** (**A**) KD or KO of UBA5 prevents the formation of charged UFC1. Western blot analysis of the levels of UBA5, UBA5~UFM1, UFC1, and UFC1~UFM1 in the indicated cells. Only in the absence of BME were the charged forms of UBA5 or UFC1 detected. (**B**) OE of EGFP-UBA5 wildtype but not C250A reduces the level of charged UFC1. Western blot analysis of HeLa and HEK293T cells that express the indicated genes. Levels of EGFP-UBA5 were detected using fluorescent gel. (**C**,**D**) Quantification of the level of UFC1~UFM1 in HeLa cells or HEK 293T that express the indicated genes. Each value represents the mean ± SD. The *p* value was determined by Student’s *t* test. (**E**) Co-expression of UBA5 with UFM1 rescues the level of UFC1~UFM1. Western blot analysis for the indicated proteins. For UFM1 overexpression we used UFM1-EGFP fusion. This enforces cleavage of EGFP from UFM1 to expose the C-terminal Gly of UFM1 for conjugation. In this Figure we show the level of UFM1 after cleavage from EGFP (See Appendix A for comprehensive analysis of UFM1-EGFP expression). Endogenous UBA5 and UBA5~UFM1 are indicated. EGFP-UBA5 runs in two forms in the SDS-PAGE, neither of which are sensitive to BME. The charged form of EGFP-UBA5 overlaps with the two bands of EGFP-UBA5 (see Appendix A for the detection of EGFP-UBA5~UFM1 using anti-UFM1 antibodies). As expected, the charged form is sensitive to BME. (**F**) Representative image of an in vitro UFC1 charging assay using 1 µM of UFC1 and UFM1 together with varying concentrations of UBA5 (indicated). Samples were loaded on SDS-PAGE and stained with InstantBlue Coomassie Protein Stain (see Appendix A for the purity of UBA5 and the band corresponding to charged UBA5). (**G**) Quantification of uncharged UFC1 (30 min) relative to its total level (time 0). Each value represents the mean ± SD.

**Figure 3 ijms-23-07445-f003:**
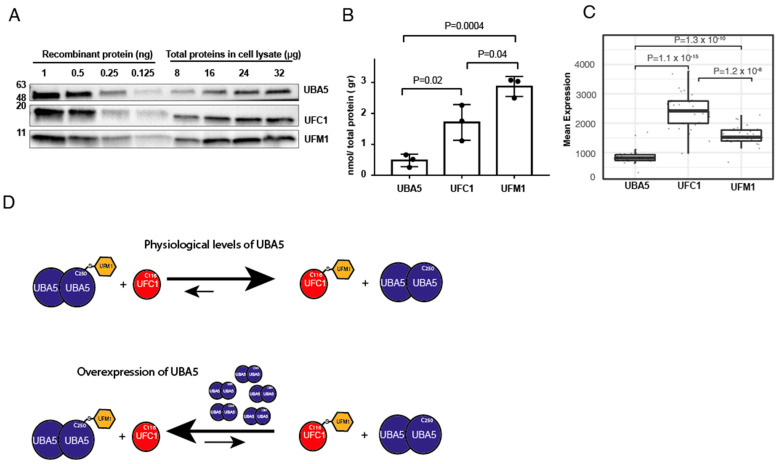
**Cells express less UBA5 than UFC1 or UFM1.** (**A**) Representative western blot analysis for the expression level of the indicated proteins in HEK293T cells. In order to translate band intensity for amount of protein, known amounts of the indicated proteins were used to generate a calibration curve (see method section for details). (**B**) Quantification of the level of indicated proteins in HEK293T cell lysate. Each value represents the mean ± SD. The *p* value was determined by Student’s *t* test. (**C**) Box plot showing the mean RNA expression level of indicated genes from different tissues (see Appendix A). Data obtained from cBioPortal and analyzed as described in method section. (**D**) Model showing the effect of UBA5 expression level on the level of charged UFC1. At physiological levels of UBA5, charged UFC1 is accumulated. However, in the case of UBA5 overexpression, the reverse reaction is preferred, and accordingly the level of charged UFC1 decreases.

## Data Availability

Not applicable.

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
