# Peer review of "Overexpression of UBA5 in Cells Mimics the Phenotype of Cells Lacking UBA5"

_ijms, 2022, doi:10.3390/ijms23137445_

Round 1

Reviewer 1 Report

The report by Sujata Kumari and co-workers entitled „ Overexpression of UBA5 in cells mimics the phenotype of cells 2 lacking UBA5” presents data showing that overexpression of the E1 protein UBA5 interferes with the transfer of the small protein UFM1 onto an as yet unknown target protein. This impairment manifests itself as a cell migration defect. Co-overexpression of UBA4 and UFM1 mitigates this defect again.

While the data are consistent and in support of the main conclusion, I do have a few comments to improve the manuscript.

 Comments:

L42-43 the sentence starting “In contrast, most UBLs… is unclear

L44 that was shown recently

L57-59: While I do understand that the mechanistic link between cell migration and Ufm1 is not yet known, the authors must have had a reason to use cell migration as the endpoint for their study. Please explain to the reader why cell migration was tested to assay the functionality of the UBA4 pathway.

L83: EGFP-UBA5 with a mutation in its active site (C250A)

Fig 1 B please label the bands

Fig 1C Although the Western Blot signal of UBA5 in the HeLa Uba5-KD cell line is gone, there is still some residual cell migration activity in this cell line (there is however none in the Ufc1-KD cell line). If UBA5 were the only E1 as stated in L209, there is either some residual Uba5 activity left or another E1 may be able to partly compensate for the loss of the Uba5 activity. Please comment on this in the manuscript.

L92 please explain to the reader what BME is and does in this experimental context

Fig 2B please label the 25kDa double band with EGFP

Fig 2E please explain to the reader what the UBA5 multiple bands are which are insensitive to BME. Please label the Ufc1-Ufm1 band that is BME sensitive

Fig 2F Is the decrease in charged Ufc1-Ufm1 in the response to an increase in UBA5 ATP-dependent? What happens when the same experiment is done in the absence of ATP or when an access of an ATP analogue is added? Are the high molecular bands of Uba5 that increase with its increasing concentration Ufm1 charged Uba5 forms? Please show also a Western blot of the in vitro experiment probed for Uba5, Ufc1 and Ufm1.

Discussion:

What about overexpression of the E1 enzyme(s) for Ubiquitin? Is it known whether it also blocks transfer or ubiquitin to the E2 or is this observation specific to the Ufm1 pathway? Please comment on this.

 Beside the reverse transfer of Ufm1 from Ufc1 to Uba5, an alternative explanation may be that overexpression of Uba5 titrates out Ufm1. Hence, less charged Ufc1 would be observed. This could also be complemented by the co-overexpression of Ufm1. Please discuss.

Author Response

The report by Sujata Kumari and co-workers entitled „ Overexpression of UBA5 in cells mimics the phenotype of cells 2 lacking UBA5” presents data showing that overexpression of the E1 protein UBA5 interferes with the transfer of the small protein UFM1 onto an as yet unknown target protein. This impairment manifests itself as a cell migration defect. Co-overexpression of UBA4 and UFM1 mitigates this defect again.

While the data are consistent and in support of the main conclusion, I do have a few comments to improve the manuscript.

We would like to thank the reviewer for his/her comments that have helped us to significantly improve our manuscript.

 Comments:

L42-43 the sentence starting “In contrast, most UBLs… is unclear

In the revised manuscript we modified the sentence (L42-44).

L44 that was shown recently

We corrected the sentence (L45)

L57-59: While I do understand that the mechanistic link between cell migration and Ufm1 is not yet known, the authors must have had a reason to use cell migration as the endpoint for their study. Please explain to the reader why cell migration was tested to assay the functionality of the UBA4 pathway.

We have added a sentence (with references) that connects ufmylation and cell migration (L59-60). 

L83: EGFP-UBA5 with a mutation in its active site (C250A)

Done (L86)

Fig 1 B please label the bands

Done

Fig 1C Although the Western Blot signal of UBA5 in the HeLa Uba5-KD cell line is gone, there is still some residual cell migration activity in this cell line (there is however none in the Ufc1-KD cell line). If UBA5 were the only E1 as stated in L209, there is either some residual Uba5 activity left or another E1 may be able to partly compensate for the loss of the Uba5 activity. Please comment on this in the manuscript.

We followed the reviewer’s suggestion and in the revised version we discuss this point (L81-83), which is supported by our in vitro gel (Fig. 2F) where we can hardly see the UBA5 band (lowest concentration), but the charging UFC1 was still possible. Therefore, we conclude that in the UBA5 KD cells, the small amount of residual UBA5, while not visible in the western blot, still allows a certain level of ufmylation.

L92 please explain to the reader what BME is and does in this experimental context

We added in the figure legend an explanation about BME and its effect on thioester bonds (L96-98).

Fig 2B please label the 25kDa double band with EGFP

Done

Fig 2E please explain to the reader what the UBA5 multiple bands are which are insensitive to BME. Please label the Ufc1-Ufm1 band that is BME sensitive

We labeled the different forms of UBA5 in the figure. In the legend, we explained that EGFP-UBA5 has two bands in the gel (L152-155). To see the charged form of EGFP-UBA5~UFM1 we referred to sup. Fig. 1 where this form is detected using anti-UFM1 antibodies. This form, as expected, is sensitive to BME.

Fig 2F Is the decrease in charged Ufc1-Ufm1 in the response to an increase in UBA5 ATP-dependent? What happens when the same experiment is done in the absence of ATP or when an access of an ATP analogue is added? Are the high molecular bands of Uba5 that increase with its increasing concentration Ufm1 charged Uba5 forms? Please show also a Western blot of the in vitro experiment probed for Uba5, Ufc1 and Ufm1.

We would like to thank the reviewer for raising this point. Our zero-time point is taken before ATP was added, and as expected, charged UFC1 is not observed, suggesting that ATP is required. However, a band that looks like the charged UBA5 starts to appear at the zero time point for reactions with UBA5 > 1 mM. To define the nature of the additional UBA5 bands, we added a gel showing UBA5 alone or together with UFM1 but without ATP (Sup Fig. 2). The gel clearly shows that the large molecular weight bands are contaminations that came from the UBA5 preparation. However, an additional band appears only in the samples containing UFM1, suggesting charging of UBA5 at time zero. This charging is due to the ATP that comes from the E. coli and is retained during the purification steps. Therefore, although ATP is not added in the reaction, the bound ATP that comes with UBA5 allows the charging of UBA5 with UFM1. Of note, since the ratio of ATP to UBA5 is 1:1, the concentration of ATP cannot exceed the concentration of UBA5.

Discussion:

What about overexpression of the E1 enzyme(s) for Ubiquitin? Is it known whether it also blocks transfer or ubiquitin to the E2 or is this observation specific to the Ufm1 pathway? Please comment on this.
This is a good question for which we still do not have an answer. In the discussion, we refer to this point (L218-224) and suggest that in the case of UBA1 overexpression, the interpretation is more complex due to the high number of E2s that function with this enzyme.

Beside the reverse transfer of Ufm1 from Ufc1 to Uba5, an alternative explanation may be that overexpression of Uba5 titrates out Ufm1. Hence, less charged Ufc1 would be observed. This could also be complemented by the co-overexpression of Ufm1. Please discuss.

This is an interesting point. However, our observation that only overexpression of UBA5 WT, and not catalytically inactive C250A, reduces the level of UFC1~UFM1, implies that the formation of UBA5 ~UFM1 is a critical step. UBA5~UFM1 is potent at transferring UFM1 to UFC1, thereby we assume that the reduction in charged UFC1 is not due to the inability of UBA5 to transfer UFM1 to UFC1.

Reviewer 2 Report

For authors

Reviewing the manuscript entitled, “Overexpression of UBA5 in cells mimics the phenotype of cells lacking UBA5” by Kumari S et al., this is an article focusing on importance of optimum expression level in UBLs. The authors explained the importance of optimal expression of UBLs by combining the loss of function experiments and the gain of function experiments. The authors need to respond to the following concerns.

 The authors need to add the results using 293T cells and its gene modified cells in Figure C, D, E. Otherwise, the authors need to describe the reason for showing only the results of Hela cells and its gene modified cells.

 The authors need to modify Figure 2E. Is UBA5-UFM1 missing?

 The authors need to modify Figure 3D. You mentioned “High levels of UBA5” in Figure 3D. You need to change it to a figure that we can understand.

Author Response

Reviewing the manuscript entitled, “Overexpression of UBA5 in cells mimics the phenotype of cells lacking UBA5” by Kumari S et al., this is an article focusing on importance of optimum expression level in UBLs. The authors explained the importance of optimal expression of UBLs by combining the loss of function experiments and the gain of function experiments. The authors need to respond to the following concerns.

We would like to thank the reviewer for his/her comments that helped us to significantly improve our manuscript.

The authors need to add the results using 293T cells and its gene modified cells in Figure C, D, E. Otherwise, the authors need to describe the reason for showing only the results of Hela cells and its gene modified cells.

We performed the trans-well experiment with HeLa cells and not HEK293T because: 1) they are larger in size which facilitates their counting; and 2) they are not easily detached from the seeded chamber during the washing steps. We have added these reasons in the text (L289-291).

 The authors need to modify Figure 2E. Is UBA5-UFM1 missing?

We agree with the reviewer that the charged UBA5~UFM1 is not easily detected in this figure. In the revised manuscript we labeled the different forms of UBA5 in Fig. 2E. In the gel, EGFP-UBA5 runs as two bands. These bands overlap with the band corresponding to EGFP-UBA5~UFM1. Therefore, in the revised manuscript, we refer to Sup Fig. 1 where we used anti UFM1 to detect the charged EGFP-UBA5. In this supplementary figure the EGFP-UBA5~UFM1 is clearly observed in the absence of BME and, as expected, disappears in the presence of BME.

The authors need to modify Figure 3D. You mentioned “High levels of UBA5” in Figure 3D. You need to change it to a figure that we can understand.

We followed the reviewer's comment and modified the model in Figure 3D.  

Round 2

Reviewer 1 Report

Dear authors,

thank you very much for your constructive replies to my comments. I would have only two final requestes.

Number 1

Please include the key message of

"We would like to thank the reviewer for raising this point. Our zero-time point is taken before ATP was added, and as expected, charged UFC1 is not observed, suggesting that ATP is required. However, a band that looks like the charged UBA5 starts to appear at the zero time point for reactions with UBA5 > 1 mM. To define the nature of the additional UBA5 bands, we added a gel showing UBA5 alone or together with UFM1 but without ATP (Sup Fig. 2). The gel clearly shows that the large molecular weight bands are contaminations that came from the UBA5 preparation. However, an additional band appears only in the samples containing UFM1, suggesting charging of UBA5 at time zero. This charging is due to the ATP that comes from the E. coli and is retained during the purification steps. Therefore, although ATP is not added in the reaction, the bound ATP that comes with UBA5 allows the charging of UBA5 with UFM1. Of note, since the ratio of ATP to UBA5 is 1:1, the concentration of ATP cannot exceed the concentration of UBA5."

in the manuscript between L127 - 137 including a reference to Sup Fig. 2

Number 2

please include the alternative expanation of the titrating out and the reasons why you think it is a less likely explanation in the Diskussion

"This is an interesting point. However, our observation that only overexpression of UBA5 WT, and not catalytically inactive C250A, reduces the level of UFC1~UFM1, implies that the formation of UBA5 ~UFM1 is a critical step. UBA5~UFM1 is potent at transferring UFM1 to UFC1, thereby we assume that the reduction in charged UFC1 is not due to the inability of UBA5 to transfer UFM1 to UFC1."

Author Response

In the revised manuscript we included the key message and referred to Sup Fig. 2 ( L136-141)

The alternative explanation is now included in the discussion ( L201-204)